# A Systematic Review and Meta-Analysis on the Impact of Statin Treatment in HIV Patients on Antiretroviral Therapy

**DOI:** 10.3390/ijerph20095668

**Published:** 2023-04-27

**Authors:** Kabelo Mokgalaboni, Wendy Nokhwezi Phoswa, Samantha Yates, Sogolo Lucky Lebelo, Sphiwe Madiba, Perpetua Modjadji

**Affiliations:** 1Department of Life and Consumer Sciences, College of Agriculture and Environmental Sciences, University of South Africa, Florida Campus, Johannesburg 1709, South Africa; phoswwn@unisa.ac.za (W.N.P.); blakes@unisa.ac.za (S.Y.); lebelol@unisa.ac.za (S.L.L.); 2Faculty of Health Sciences, University of Limpopo, Polokwane 0700, South Africa; sphiwe.madiba@ul.ac.za; 3Non-Communicable Disease Research Unit, South African Medical Research Council, Cape Town 7505, South Africa; perpetua.modjadji@mrc.ac.za

**Keywords:** HIV, statin, antiretroviral therapy, CD4 count, immune activation, viremia

## Abstract

The rate of new human immunodeficiency virus (HIV) infections globally is alarming. Although antiretroviral therapy (ART) improves the quality of life among this group of patients, ARTs are associated with risk of cardiovascular diseases (CVD). Moreover, virally suppressed patients still experience immune activation associated with HIV migration from reservoir sites. Statins are widely recommended as therapeutic agents to control ART-related CVD; however, their impacts on the cluster of differentiation (CD)4 count and viral load are inconsistent. To assess the effect of statins on markers of HIV infections, immune activation and cholesterol, we thoroughly reviewed evidence from randomised controlled trials. We found 20 relevant trials from three databases with 1802 people living with HIV (PLHIV) on statin–placebo treatment. Our evidence showed no significant effect on CD4 T-cell count standardised mean difference (SMD): (−0.59, 95% confidence intervals (CI): (−1.38, 0.19), *p* = 0.14) following statin intervention in PLHIV on ART. We also found no significant difference in baseline CD4 T-cell count (SD: (−0.01, 95%CI: (−0.25, 0.23), *p* = 0.95). Our findings revealed no significant association between statins and risk of viral rebound in PLHIV with undetectable viral load risk ratio (RR): (1.01, 95% CI: (0.98, 1.04), *p* = 0.65). Additionally, we found a significant increase in CD8^+^CD38^+^HLA-DR^+^ T-cells (SMD (1.10, 95% CI: (0.93, 1.28), *p* < 0.00001) and CD4^+^CD38^+^HLA-DR^+^ T-cells (SMD (0.92, 95% CI: (0.32, 1.52), *p* = 0.003). Finally, compared to placebo, statins significantly reduced total cholesterol (SMD: (−2.87, 95% CI: (−4.08, −1.65), *p* < 0.0001)). Our results suggest that the statin lipid-lowering effect in PLHIV on ART may elevate immune activation without influencing the viral load and CD4 count. However, due to the limited evidence synthesised in this meta-analysis, we recommend that future powered trials with sufficient sample sizes evaluate statins’ effect on CD4 count and viral load, especially in virally suppressed patients.

## 1. Introduction

Human immunodeficiency virus (HIV) is a public health concern; according to the 2021 UNAIDS report, about 38.4 million people were living with HIV globally and 1.5 million new infections and 650,000 deaths from AIDS-related illnesses were recorded [1]. About 7.5 million children and adults in South Africa were living with HIV in 2021, with 210,000 new infections and 51,000 AIDS-related deaths reported [2]. This significant rate of new infections suggests an increasing need to combat this virus through prevention and treatment, respectively. This virus attacks the individual immune system, also known as cluster of differentiation 4 (CD4) counts, and pre-disposes infected individuals to various infections, including metabolic and acquired immune deficiency virus syndrome (AIDS) [3]. HIV has received enormous research attention since its outbreak; most recently, researchers have been exploring the bone marrow transplantation approach, which seems to be associated with the elimination of HIV reservoirs [4]. However, this is risky as it involves replacing individual bone marrow with HIV-resistant stem cells from a donor, implying that the patients will be susceptible to opportunistic infections during this procedure.

Nevertheless, the introduction and progress made with antiretroviral therapy (ART) have improved the quality of life for people living with HIV (PLHIV); the number of AIDS-related complications and deaths has since decreased in those that adhere to ART [5,6]. The fundamental goal of ART is to improve the immune system through increasing CD4 count and successfully suppressing HIV. This method stops the disease’s progression and reduces the risk and vulnerability to opportunistic infections. However, ART has reportedly been linked to an increased risk of cardiovascular disease (CVD) [7], as demonstrated by impaired lipid profile [8].

Additionally, kidney malfunctioning has been noted in other PLHIV on ART [9]. Therefore, due to the above secondary complications other patients are still reluctant to take ART as they show adversity in other HIV individuals. It is worth noting that PLHIV currently on ART are prescribed statins, a lipid-lowering treatment to control cholesterol and HIV-ART-related dyslipidaemia [10]. However, their impact, especially on viral load and CD4 count in PLHIV on ART, is unclear. This problem has prompted other researchers to conduct clinical trials investigating this paradigm. Surprisingly, researchers have reported contradicting findings, with some showing consistently improved HIV surrogate markers [11,12,13,14,15,16,17,18] and others showing negative findings [19,20,21,22,23]. HIV as a chronic infection is closely associated with inflammation due to the continuously rising viral load [24,25]. Interestingly, this seems to be controlled in PLHIV who are virally suppressed [26]. Therefore, any potential pharmaceutical agent that can target inflammatory pathways may be of interest when it comes to the development of drugs that can curb inflammation in PLHIV.

Although the effect of statins on inflammation is well documented in diabetes [27], this aspect is not clear in PLHIV who are prescribed ART [12,16,20,21]. We believe that amelioration of inflammation by statins may partly contribute to reducing the risk of CVD and, thus, improving the overall quality of health. Disappointingly, Nachega et al., 2012 [28] have highlighted a serious concern regarding the administration of ART with polypharmacy as this is associated with drug–drug interactions. This subsequently results in drug toxicity, poorer ART adherence, and loss of efficacy of the used ART in older PLHIV. The same sentiments were expressed by Potentelo et al. [29]. Therefore, all these concerns make us question whether statins are safe for PLHIV taking ART. We have noted a few literature reviews that evaluated data on the beneficial effects of statins against secondary complications in PLHIV [30,31,32]; however, the impact on surrogate markers of HIV infections and cholesterol is inconsistent. Therefore, the current meta-analysis sought to evaluate whether statin treatment in HIV patients improves or worsens the surrogate markers of HIV infection, focusing on CD4 count, viral load and markers of immune activation. Furthermore, we sought to evaluate the effect of statin treatment on cholesterol in virally suppressed patients. 

## 2. Materials and Methods

### 2.1. Study Design and Registration

The current study was conducted and reported following the formatting guidelines recommended by preferred reporting items for systematic reviews and meta-analysis (PRISMA) guidelines [33]. Due to the study design, no institutional ethics approval and patient informed consent were required. Only evidence from randomised controlled trials (RCTs) was sought and analysed. The protocol that accompanies this review has been registered with the PROSPERO registry: CRD42023397964.

### 2.2. Search Strategy and Information Sources

Relevant RCTs were searched from inception until February 2023 and retrieved from MEDLINE, Scopus, and Cochrane Library, irrespective of the language, using the medical subject headings (MesH) terms or text words. The following search terms were applied: “statin” and synonyms such as “hydroxyl glutaryl (HMG-CoA) reductase inhibitors”, “atorvastatin”, “rosuvastatin”, “fluvastatin”, “lovastatin”, “pitavastatin”, “simvastatin” using OR as a Boolean AND “HIV” OR “AIDS” OR “human AND immunodeficiency” AND “Antiretroviral therapy” OR “ART”. Exact search strategies adapted to these databases are presented in Appendix A.

### 2.3. Eligibility Criteria and Study Selection

All records obtained from these databases were exported to Mendeley Desktop software and duplicate records were excluded. Two researchers (K.M. and W.N.P.) screened the remaining records independently and disagreements were resolved by consensus. Retrieved full-text articles were thoroughly evaluated for eligibility and subjected to meta-analysis if they met all the following inclusion criteria: (i) human studies; (ii) PLHIV on ART; and (iii) treated with any form of statin. In contrast, the following types of studies were all excluded: (i) animal models; (ii) without control; (iii) comparative trials; (iv) PLHIV not on ART or ART treated PLHIV not on statin; and (v) studies not reporting any of the primary markers of HIV infections and cholesterol. 

### 2.4. Data Extraction and Quality Assessment

Two researchers (K.M. and W.N.P.) independently extracted information from relevant studies using a pre-defined data extraction form. Briefly, the following data items were extracted from each of the studies: author’s surname and year of publication; study design, population states, and sample size in each group; the country where the study was conducted; dosage form of statin and duration of intervention; age and sex of study participants; surrogate markers evaluated; and general findings. Disagreements in data extractions were resolved by discussion. K.M. and W.N.P. independently evaluated the quality of included RCTs using a modified JADAD scale [34]. This tool focuses on these four main domains: (i) description of randomisation; (ii) accuracy of double blinding; (iii) description of patient loss or exclusion; and (iv) adequacy of randomisation. Any disagreement was resolved by inviting the third researcher, P.M. and re-evaluating the study and domains in dispute.

### 2.5. Data Synthesis and Analysis

All analyses were conducted using RevMan 5.4 software (Review Manager (RevMan), version 5.4.1, The Cochrane Collaboration, 2020) and metaHUN, a web tool accessible at http://softmed.hacettepe.edu.tr/metaHUN/ (accessed on 24 February 2023). Effect size from all outcomes for continuous data was estimated by computing each study’s mean, standard deviation (SD), and sample size. Mean and SD were estimated following guidelines [29] in cases where the included study reported only mean and interquartile range (IQR); when the standard error of the mean (SER) was given, SD was estimated using SD = SER × √n [35]. For other outcomes, we extracted or estimated a change in mean and SD from baseline and post-test results using the equation (∆mean = (m_f_ − m_b_)). We further estimated change in SD using the following formula: $\Delta \mathrm{SD}=\sqrt{{\left({\mathrm{SD}}_{b}\right)}^{2}+{\left({\mathrm{SD}}_{f}\right)}^{2}-2(r\times {\mathrm{SD}}_{b}\times {\mathrm{SD}}_{f}});$ this formula was outlined by Cochrane guidelines and other researchers’ findings, with a correlation coefficient (r) of 0.7 used in both groups [36,37]. Statistical heterogeneity was estimated using the I^2^ and Cochran Q tests [38]. I^2^ values below 50% and above 75% were classified as minimal and substantial statistical heterogeneity, respectively. Random-effect and fixed models were used in the presence and absence of statistical heterogeneity, respectively. We performed sub-group analyses to determine the sources of clinical heterogeneity across studies [39]. Our sub-group analyses explored the impact of statin and statin doses. The funnel plots were used to assess publication bias. We further conducted a sensitivity analysis to evaluate the stability of our result [40]. 

## 3. Results

### 3.1. Search, General Characteristics and Quality of Included Trials

The comprehensive search for the literature from these databases yielded 795 records that were screened and evaluated at the title, abstract and keywords level (Figure 1). Finally, 20 studies [12,13,14,16,19,20,21,22,23,41,42,43,44,45,46,47,48,49,50,51,52] were included, of which 12 trials used rosuvastatin of either 10, 20 or 80 mg, four used atorvastatin of 10, 30 and 40 mg, four used pravastatin at 40 mg and one used pitavastatin of 2 mg. These studies were published in peer-reviewed journals between 2004 and 2021, thus tracing old and advanced evidence on statins in PLHIV taking ART. Of 1802 PLHIV taking ART, 953 people taking a statin and 871 taking a placebo treatment were included. About 760 (80%) participants on statin treatment were males. The overall mean and SD ages of participants on statin treatment were 47.48 ± 4.96 years (Table 1). Publications of these trials were distributed across seven countries. Briefly, 12 studies [14,22,23,42,43,45,46,47,49,50,52,53] were conducted in the United States of America, three studies [12,21,44] were conducted in Australia, two [16,20] were conducted in Switzerland, and one study was conducted in each of the following countries, China [19], France [41] and Uganda [48]. For all four domains, we found the quality to be good: domain one received an overall score of 20 out of a possible score of 21, domain two scored 18 out of 21, and domains three and four received at least 14 out of 21. Across individual studies, about two were rated fair as they scored at least two points, ten rated good with a score of about three and, lastly, eight studies scored four points and were rated excellent (Appendix A).

### 3.2. Effect of Statin on Markers of HIV Infection

#### 3.2.1. The Effect of Statin on Baseline CD4 T-Cell Count in HIV Patients on Antiretroviral Therapy (ART)

All included trials [12,13,14,16,19,20,21,22,23,41,42,43,44,45,46,47,48,49,50,51,52] investigated baseline CD4 counts in PLHIV taking both ART and statin treatments. Following our meta-analysis of the collected data, we found no significant difference in CD4 counts between PLHIV taking a statin and those taking a placebo, with a standardised mean difference (SMD) (−0.01), 95% CI: (−0.25, 0.23), *p* = 0.95). Notably, statistical heterogeneity was high (I^2^ = 83%) (Appendix A). 

We also collected data on CD4 T-cell counts to find the effect of statins following the intervention period. Only three trials [12,20,41] reported sufficient data on CD4 T-cells or data to estimate the change in CD4 T-cell counts. The overall pooled effect estimates showed comparable results between PLHIV taking a statin and those taking a placebo (SMD (−0.59), 95% CI: (−1.38, 0.19), *p* = 0.14). Similarly, there was a high statistical heterogeneity (I^2^ = 72%) among the analysed evidence (Figure 2). We further assessed the magnitude of the effect of statin and found that, according to Cohens d interpretation, the statin in PLHIV on ART might have a medium suppressive effect (SMD > 0.5) on CD4 T-cell count.

#### 3.2.2. Effect of Statin on CD8^+^, and CD4^+^CD38^+^HLA-DR^+^ Cells

We evaluated the effect of statins on markers of immune activation. Evidence from five trials revealed a significant increase in CD8^+^CD38^+^HLA-DR^+^ T-cells in PLHIV taking ART who were also treated with a statin (SMD (1.10), 95%CI: (0.93, 1.28), *p* < 0.00001). Interestingly, no evidence of heterogeneity was observed (I^2^ = 0%) (Figure 3A). Similarly, there was a significant increase in CD4^+^CD38^+^HLA-DR^+^ T-cells (SMD (0.92), 95% CI: (0.32, 1.52), *p* = 0.003) (Figure 3B).

#### 3.2.3. Effect of Statin on Undetectable HIV Load (<50 HIV RNA Copy)

Sixteen RCTs [13,14,16,19,21,22,23,42,44,45,46,47,49,50,51,52] evaluated the effect of statin on undetectable viral copies in HIV patients taking ART. Our overall effect estimates showed a non-significant risk of experiencing a viral rebound in PLHIV-ART with an undetectable viral load on statin treatment compared to placebo, with a risk ratio (RR) of (1.01, 95% CI: (0.98, 1.04), *p* = 0.65). This suggests that statin treatment in PLHIV with ART-suppressed viremia does not present a risk of viral duplication. Interestingly, there was no evidence of heterogeneity (I^2^ = 0%) among the analysed studies (Appendix A).

### 3.3. Effect of Statin on Total Cholesterol in HIV Patients on ART

Total cholesterol (TC) data were extracted from nine studies [12,16,19,20,41,42,44,47,50] with 200 PLHIV taking statin treatments. The pooled effect estimates revealed a significant reduction in TC following statin treatment compared to placebo treatment (SMD (−2.87), 95% CI: (−4.08, −1.65), *p* < 0.00001) (Figure 4). However, we observed substantial heterogeneity amongst these studies (I^2^ = 95%).

### 3.4. Test for Sub-Group Differences

Due to substantial statistical heterogeneity observed in our effect estimates, we conducted a sub-group analysis to explore the sources of heterogeneity based on the doses and form of statins across all outcomes. Data supporting these are presented in a Appendix A. Briefly, a test for sub-grouping differences according to the form of statin on baseline CD4 T-cell count showed an (SMD (−0.01), 95% CI:(−0.25, 0.21), I^2^ = 0%) (Appendix A). This suggests that the observed heterogeneity may arise from the use of different forms of statin treatment. Due to the presence of only three studies, no sub-group was conducted on change in CD4 T-cell counts and CD4^+^CD38^+^HLA-DR^+^ T-cells. A sub-group difference according to statin doses on TC revealed an (SMD (−3.06), 95% CI: (−4.15, −1.98), I^2^ = 0%) (Appendix A), while 40 mg of statin showed a pronounced decrease in TC (SMD (3.48), 95% CI: (−5.35, −1.62), I^2^ = 93%. Consistently, a decrease in TC was observed when the sub-group was performed according to the form of statin (SMD (−2.73) 95% CI: (−3.77, −1.70), I^2^ = 40.5%) (Appendix A). This suggests that the statistical heterogeneity observed might not be due to the forms of statin used but associated with different doses used.

### 3.5. Sensitivity Analysis

Sensitivity analysis was used to evaluate the robustness of our findings and was conducted following a one-study exclusion approach. This method was based on the sample size of the study. After removing a study [41] with a small sample size, we found that the baseline CD4 T-cell count effect size changed to SMD (0.03), 95% CI: (−0.22, 0.27). However, this was statistically not significant (*p* = 0.82). For a change in CD4 T-cell count, we removed a study [20] with a large sample size and found that the effect size increased significantly SMD (−0.96), 95% CI: (−1.90, −0.03), *p* = 0.04; this finding was about 0.37 (37%) increase from the original effect size. Lastly, for TC the exclusion of the study by Bonnet et al. [41] due to the small sample size resulted in a minimal increase in TC, as demonstrated by SMD (−2.83), 95% CI: (−4.13, −1.53), which was a 5% decrease from original effect size. Removing a study by Nakanjako et al. [48] due to small sample size, our effect size changes significantly from original to CD8^+^CD38^+^HLA-DR^+^T-cells SMD (1.14), 95% CI (0.96, 1.32), *p* < 0.00001 and CD4^+^CD38^+^HLA-DR^+^ T-cell SMD (0.70), 95% CI: (0.10, 1.29), *p* = 0.02.

### 3.6. Publication Bias

We also evaluated publications using visual inspection of funnel plots and QQ-normal plots. At least four studies provided sufficient data about the change in the CD4 T-cell count level, while visually the funnel plot revealed a symmetrical shape, suggesting no evidence of publication bias (Figure 5A). Additionally, our Q-Q plot also showed that all studies were closer to the straight line, thus suggesting no publication bias (Figure 5B). On the other hand, we noticed evidence of publication bias across studies that reported on the level of total cholesterol, as indicated by an asymmetrically shaped funnel plot coupled with Egger’s regression test (Q = 22.3847, *p* = 0.0000) (Appendix A).

## 4. Discussion

According to our knowledge, no meta-analysis explored the effect of statin on an ART-enhanced CD4 T-cell count, undetectable HIV-RNA, and markers of T-cell activation amongst ART-treated PLHIV co-supplemented with statins. In this study, we have explored the RCT evidence investigating the impact of statin on the CD4 T-cell count, undetectable HIV-RNA copy, markers of T-cell activation and TC level in PLHIV on ART. We found that in about 1802 PLHIV on ART, there were no significant effects of statin on either baseline or post-treatment CD4 T-cell count compared to placebo. We also found a significant elevation in markers of T-cell activation, with a focus on both CD8^+^ CD38^+^ and CD4^+^CD38^+^HLA-DR^+^ T-cells. Furthermore, we showed that statins in PLHIV treated with ART significantly reduce TC levels. 

Although we had many trials in this analysis, pooled evidence from three studies with sufficient data on post-treatment CD4 T-cell count suggests that statin does not significantly impact CD4 T-cell count in virally suppressed patients. Our findings are supported by results obtained from previous trials that reported no significant increase in CD4 T-cell count following statin treatment in PLHIV on ART [22,48]. In contrast to our overall findings, a study by (Rodriguez et al., 2007) reported reduced CD4 T-cell gain following statin treatment in viremic-suppressed patients [11]. Long-term ART in HIV is associated with improved CD4 count; however, this depends on the level of CD4 count at the time of ART initiation, with low CD4 count failing to normalise even after prolonged ART administration [54]. On the other hand, in PLHIV who are ART-naïve rosuvastatin administration also revealed no significant increase in CD4 T-cell count [55]. This overall trend in CD4 T-cell counts suggests that statin may not effectively offer beneficial effects in improving CD4 count, especially in virally suppressed patients. Statins are known to have anti-inflammatory properties, while it has been reported that rosuvastatin significantly reduces markers of inflammation in HIV patients on ART [22]. Hearps et al. [56] support the sentiments of Funderberg [22]; however, they reported no normalisation of CD4 T-cell counts in the virally suppressed patients despite amelioration of inflammation. On the other hand, results from ART naïve PLHIV indicate that low serum cholesterol in children and adults is associated with reduced production of interleukin-2 and CD4 T-cell counts [57,58]. This suggests that the cholesterol-lowering effect of these statin therapies may induce CD4 T-cell reduction.

As demonstrated by a comparable level of suppressed viral load, our findings suggest that statin administration in virally suppressed patients is not associated with the risk of viral rebound. These results are corroborated by other researchers who have shown no effect of statin on viral load in HIV on stable ART [19,47]. In disagreement, Do et al. [52] reported that about 81% of PLHIV with a viral load below 50 maintained the undetectable state compared to 83% on placebo. Similarly, 87% and 92% were reported to have a viral load below 50 copies in statin and placebo, respectively [16]. PLHIV on ART with undetectable HIV-RNA copies sometimes experience viral blips; this generally reflects on the failures of the ART if the blips continue and the viral load is consecutively above 50 HIV-RNA copies [59]. These viral blips are reportedly associated with high levels of baseline viral load prior to initiation of ART [60]. 

One reason for a continuous viral rebound in PLHIV taking ART is an elevated immune activation associated with the continuous production of viral particles from reservoir sites [61]. Our results revealed increased immune activation among virally suppressed patients on statin treatment. This pre-disposes patients to secondary complications and adversity. Previous evidence has revealed that even in successful ART functionality, demonstrated via low viral load and increased CD4 T-cell count, HIV in the reservoir still induces immune activation [62]. This partly explains increased immune activation in our current analysis, as demonstrated via increased presence of CD4^+^CD38^+^HLA-DR^+^ and CD8^+^CD38^+^HLA-DR^+^ T-cells. According to Overton et al., statins can reduce markers of immune activation, which are CD38 on CD8^+^ T cells in PLHIV on ART [63]. Immune activation is a chronic inflammatory state often observed in virally suppressed patients. One of the mechanisms by which HIV reservoirs cause immune activation is through the activation of glycoprotein 120 (gp120), which promotes the release of IL-1β from resident macrophages by binding to the chemokine receptor CCR5 [64]. On the other hand, HIV-1, via its proteins such as p17, p24, and gp41, act as a viral pathogen-associated molecular pattern (PAMP) that signals through toll-like receptor-2 (TLR2) and further promotes immunological activation via the nuclear factor-kappa-β (NF-κβ) signalling pathway [65].

Conversely, single-stranded (ssRNA) HIV-1 can interact with plasmacytoid dendritic cells, including TLR-7 and TLR-9, resulting in the release of interferon-alpha (IFN-α), which may contribute to persistent immunological activation [66]. Although atorvastatin has been shown to reduce immune activation in PLHIV, these results are observed in ART-naïve PLHIV [67]. These results suggest that there might be drug–drug interaction when statins are co-administered with ART in PLHIV. Therefore, any pharmacological approach that can effectively reduce immune activation in PLHIV may potentially reduce secondary complications associated with HIV. Due to the multifaceted nature of HIV, numerous pathways are implicated in the reduction in immune activation mediated by statin. For example, statins may reduce cyclo-oxygenase activity and prostaglandin synthesis [68]. They may also inhibit the activation of immune cells, including monocytes and macrophages, which contribute to inflammation [22,67,69].

Interestingly, atorvastatin and pravastatin are reportedly safe and effective for PLHIV on a protease inhibitor or non-nucleoside reverse transcriptase inhibitor-based ART. While rosuvastatin is safe at a lower dose, it should not be prescribed for PLHIV on any PI-based ART. Although there is data to support the use of pitavastatin, its pharmacokinetic features and few drug–drug interactions make it a safe agent in PLHIV. Due to potential drug interactions, side effects, and the paucity of clinical data, patients receiving ART should avoid taking fluvastatin, lovastatin, and simvastatin [70].

A significant reduction in TC levels, which is a primary biomarker of hyperlipidaemia and predictor of CVD, demonstrated the cholesterol-lowering effect of statins. PLHIV have an increased risk of ART-induced CVD, including dyslipidaemia; however, the administration of statins seems to alleviate these complications [71]. For instance, previous meta-analyses of cohorts and RCTs conducted in 2016 [30,31] showed a significant effect of statin on TC; however, these studies analysed data from ART-naïve and ART-treated PLHIV. Therefore, the combined results might not necessarily show the effect of statin on ART-enhanced immune function in PLHIV. Most importantly, our results are corroborated by evidence that shows a significant decrease in TC following statin treatment [12,16,19,20,41,42,44,47,50,72,73,74,75]. While such positive effects are acknowledged, Baker et al. reported conflicting results in 2012, finding no change in TC when pravastatin was administered to PLHIV on ART [76]. Such results potentially reflect the limitations of the statin lipid-lowering effect and a need for further investigations. The mechanism by which statins lower cholesterol is inhibiting the activity of 3-hydroxy-3-methylglutaryl co-enzyme A reductase, resulting in decreased cholesterol synthesis in the liver [77,78].

## 5. Strength and Limitations

This study is the first quantitative analysis to explore the effect of statin on CD4 T-cell count, viral load and immune activation in ART-controlled PLHIV. The evidence may be reliable as evidence was gathered from high-quality RCT across different databases. Other outcomes were analysed using evidence from a few studies with small sample sizes; therefore, caution must be made when interpreting the analysed evidence. Although heterogeneity was high, sub-group analysis was conducted to explore the source, which was suggested by evidence to be arising from different doses. Through sub-group analysis, we also established that pravastatin and pitavastatin effectively reduced TC amongst virally suppressed patients. Most importantly, the baseline and post-treatment CD4 T-cell counts were meta-analysed.

## 6. Conclusions

This meta-analysis highlights the importance of statin in PLHIV whose viral load is maintained below a detectable level using ART. Our findings revealed no significant differences in CD4 count (baseline and post-treatment), suggesting no beneficial effect of statins on these parameters. Notably, viral load and CD4 count are primarily used to monitor ART’s efficacy amongst PLHIV on ART. Moreover, our findings revealed no statistical association between statin and risk of viral rebounds. Moreover, we found that statin was significantly associated with an increased marker of immune activation, suggesting the risk of developing HIV-related complications. Nevertheless, statins effectively lowered TC among PLHIV on ART, suggesting that ART treatments did not overshadow the effect of statins. However, more trials may be necessary to ascertain the effect of statin on CD4 T-cell counts as this remains an essential biomarker that reflects ART efficacy in PLHIV.

## Figures and Tables

**Figure 1 ijerph-20-05668-f001:**
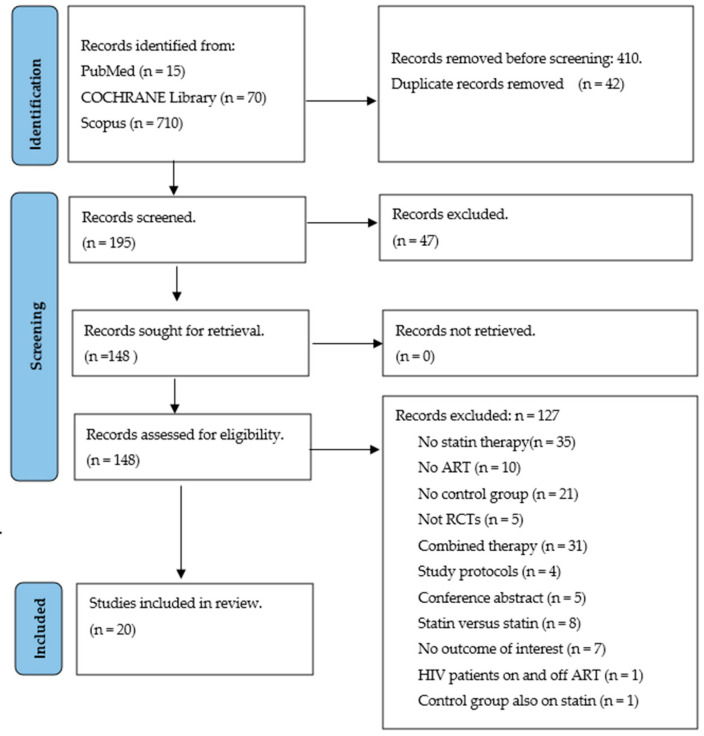
PRISMA flow diagram presenting information sources and study selection procedure followed.

**Figure 2 ijerph-20-05668-f002:**
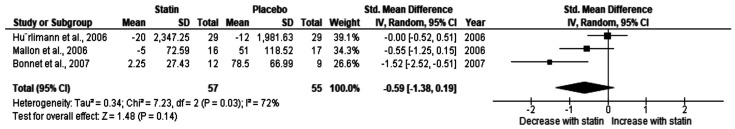
Effect of statin on follow-up CD4 T-cells in PLHIV on ART. I^2^ statistic was used to assess statistical heterogeneity. The probability value of less than 0.05 was considered statistically significant. Hürlimann et al., 2006 [20]; Mallon et al., 2006 [12]; Bonnet et al., 2007 [41].

**Figure 3 ijerph-20-05668-f003:**
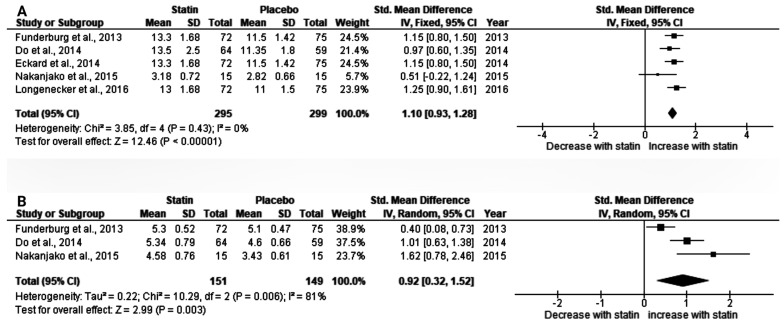
Effect of statin on markers of immune activation in PLHIV on ART. (**A**) CD8^+^ CD38+ HLA-DR^+^ T-cells; and (**B**) CD4^+^CD38^+^HLA-DR^+^ T-cells. I^2^ statistic was used to assess statistical heterogeneity. The probability value of less than 0.05 was considered statistically significant. Funderburg et al., 2013 [22]; Do et al., 2014 [52]; Eckard et al., 2014 [23]; Nakanjako et al., 2015 [48]; Longenecker et al., 2016 [21].

**Figure 4 ijerph-20-05668-f004:**
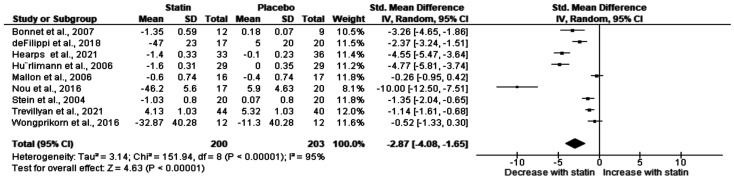
Effect of statin on total cholesterol in PLHIV on ART. I^2^ statistic was used to assess statistical heterogeneity. The probability value of less than 0.05 was considered statistically significant. Bonnet et al., 2007 [41]; Defilippi et al., 2018 [50]; Hearps et al., 2021 [16]; Hürlimann et al., 2006 [20]; Mallon et al., 2006 [12]; Nou et al., 2016 [47]; Stein et al., 2004 [42]; Trevillyan et al., 2021 [44]; Wongprikorn et al., 2016 [19].

**Figure 5 ijerph-20-05668-f005:**
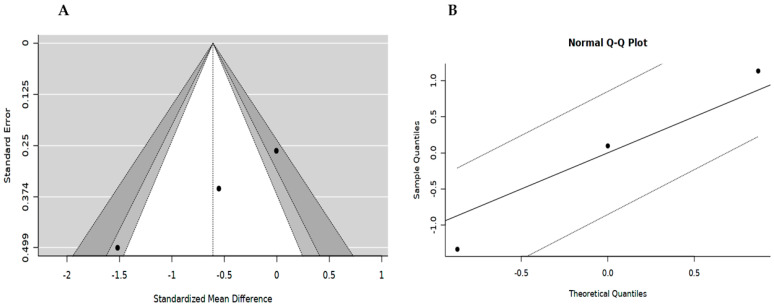
Visualisation of publication bias graphically on CD4 count. (**A**) Funnel plot; and (**B**) Q-Q normal plot.

**Table 1 ijerph-20-05668-t001:** Characteristics of included RCTs (*n* = 20) evaluating effect of statin in PLHIV on antiretroviral therapy.

Study, Year, Country	Study Design	Population Status	Duration of HIV (Years/Months)	Duration of ART (Years/Months)	Sample Size Statin Group, *n*	Form of Statin, Dosage, and Duration	Age (Years)	Gender, Male (%)
Hearps et al., 2021 [16] Switzerland	Post hoc analysis of the RCT.	Sixty-nine virologically suppressed patients.	19.1 ± 3.65	NR	33	20 mg rosuvastatin for 96 weeks.	54 ± 6.5	32 (97)
Trevillyan et al., 2021 [44] Australia	Post hoc analysis of the RCT.	Eighty-eight controlled PLHIV on ART.	17.2 ± 8.5	NR	44	20 mg rosuvastatin or 10 mg on antiretroviral therapy for 96 weeks.	53.9 ± 5.9	42 (95.5)
Kamari et al., 2019 [14] USA	Post hoc analysis of the RCT.	One hundred and forty-seven PLHIV on ART.	NR	7.0 ± 5.3	72	10 mg rosuvastatin for 96 weeks.	45 ± 9.0	58 (81)
deFillippi et al., 2018 [50] USA	Post hoc analysis of the RCT.	Forty PLHIV on stable ART.	16.8 ± 5.1	12.4 ± 3.7	19	20 mg to 40 mg atorvastatin for three months.	52.2 ± 3.8	15 (79)
Hileman et al., 2017 [49] USA	Post hoc analysis of the RCT.	One hundred and forty-seven PLHIV on ART.	11.71 ± 6.6	7.0 ± 5.3	72	10 mg rosuvastatin for 96 weeks.	45.37 ± 9.13	58 (81)
Nixon et al., 2017 [45] USA	Post hoc analysis of the RCT.	Ninety-four PLHIV on boosted PI-based ART.	6 ± 1.3	NR	94	10 mg of atorvastatin for 4 weeks, followed by 20 mg for 16 weeks.	48 ± 2.3	64 (68)
Hileman et al., 2016 [13] USA	Post hoc analysis of the RCT.	One hundred and forty-seven PLHIV on stable ART.	NR	NR	72	10 mg rosuvastatin for 48 weeks.	45.4 ± 9.1	58 (81)
Longenecker et al., 2016 [21] Australia	Post hoc analysis of the RCT.	One hundred and forty-seven PLHIV on stable ART.	11 ± 1.83	5.2 ± 1.13	72	10 mg rosuvastatin for 96 weeks.	45 ± 1.7	58 (81)
Morrison et al., 2016 [46] USA	Post hoc analysis of the RCT.	One hundred and forty-seven PLHIV on ART.	11 ± 1.83	3.9 ± 1.23	72	10 mg rosuvastatin for 24 weeks.	45 ± 1.67	58 (81)
Nou et al., 2016 [47] USA	Post hoc analysis of the RCT.	Forty PLHIV with sub-clinical coronary atherosclerosis on stable ART.	16.8 ± 5.1	12.4 ± 3.7	19	40 atorvastatin for 12 months.	52.2 ± 3.8	15 (79)
Wongprikorn et al., 2016 [19] China	Post hoc analysis of the RCT.	Twenty-four PLHIV on ART.	NR	42 ± 6.93	12	2 mg pitavastatin for 12 weeks.	49.6 ± 10.6	8 (66.7)
Do et al., 2014 [52] USA	Post hoc analysis of the RCT.	One hundred and forty-seven PLHIV on stable ART.	122 ± 31.5	56 ± 21.25	64	10 mg rosuvastatin for 96 weeks	46 ± 2.5	53 (83)
Erlandson et al., 2015 [51] USA	Post hoc analysis of the RCT.	One hundred and forty-seven PLHIV on stable ART.	NR	63 ± 13.67	72	10 mg of rosuvastatin for 96 weeks.	45.6 ± 1.72	58 (81)
Nakanjako et al., 2015 [48] Uganda	Post hoc analysis of the RCT.	Thirty PLHIV on a suppressive cART.	NR	NR	15	80 mg of rosuvastatin for 12 weeks.	43 ± 8.54	8 (53)
Eckard et al., 2014 [23] USA	Post hoc analysis of the RCT.	One hundred and forty-seven PLHIV on stable ART.	133 ± 20.7	63 ± 13.7	72	10 mg rosuvastatin 24 weeks	45.6 ± 1.72	58 (81)
Funderburg et al., 2013 [22] USA	Post hoc analysis of the RCT.	One hundred and forty-seven PLHIV on ART.	133 ± 20.7	63 ± 13.7	72	10 mg rosuvastatin for 24 weeks.	45.6 ± 1.71	58 (81)
Bonnet et al., 2007 [41] France	Post hoc analysis of the RCT.	Twenty PLHIV on stable ART.	NR	NR	12	40 mg of pravastatin for three months.	42.5 ± 2.31	11 (92)
Hürlimann et al., 2006 [20] Switzerland	Post hoc analysis of the RCT.	Twenty-nine PLHIV on a PI-containing cART.	NR	NR	29	40 mg pravastatin for 8 weeks.	43	23 (79)
Mallon et al., 2006 [12] Australia	Post hoc analysis of the RCT.	Thirty-one PLHIV with hypercholesterolaemia on PI-containing therapy.	13.5 ± 4.5	NR	16	40 mg pravastatin for 12 weeks.	52 ± 12	16 (100)
Stein et al., 2004 [42] USA	Post hoc analysis of the RCT.	Twenty PLHIV on HIV PI-based ART.	11.8 ± 0.5	8 ± 3.7	20	40 mg pravastatin for 8 weeks.	44.1 ± 1.6	18 (90)

HIV—human immune deficiency virus; RCT—randomised controlled trials; PLHIV—people living with HIV; PI—protease inhibitor; ARV—antiretroviral therapy; cART—combined antiretroviral therapy, NR—not reported.

## Data Availability

The additional information supporting this manuscript is provided as a Appendix A.

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
