# Peer review of "A Systematic Review and Meta-Analysis on the Impact of Statin Treatment in HIV Patients on Antiretroviral Therapy"

_ijerph, 2023, doi:10.3390/ijerph20095668_

Round 1

Reviewer 1 Report

The article contributes to the field of knowledge and science. The presentation of the article is excellent. However, a minor correction is required on page 3, lines 110-111. The sentence is incomplete, perhaps the author wanted to state that "In Contrast............... were excluded" Please insert "were excluded"

Author Response

Comment: The article contributes to the field of knowledge and science. The presentation of the article is excellent. However, a minor correction is required on page 3, lines 110-111. The sentence is incomplete, perhaps the author wanted to state that "In Contrast............... were excluded" Please insert "were excluded"

Response:  Thank you very much for your precious time reviewing our manuscript. As a result, we have included the phrase “were all excluded”. This is highlighted in red on page 3., line 117. This has improved the quality of our reporting in the revised manuscript version.

Reviewer 2 Report

this is a systematic review and metanalysis on statin use in HIV patients.

table 1: "form of stain" in th eheading should be corrected

it is not correct to state that RCT trials were included is the analysis. main of them were not randomized for the use of statins. in this case, they should considered and evaluated as observational studies

See ref 41 as an example: From the abstract of the study: "Forty-two PLWH aged ≥40 years receiving a protease inhibitor (PI)-based regimen were randomized (1:1) to switch from PI to Raltegravir (n = 20), or to remain on PI (n = 22). After 24 weeks, all patients received atorvastatin 20mg/day for 48 weeks."

the analysis ". Effect of statin on total cholesterol in HIV patients in ART." is difficult to understand. Statins are very well knwon to reduce cholesterol. what does this analysis add to current knowledge?

the interesting analysis is the one on CD8 and CD4 but it is performed on very few studies.

reasons for statin administration in each study is not reported

Author Response

This is a systematic review and metanalysis on statin use in HIV patients.

Table 1: "form of stain" in th eheading should be corrected

Response: Thanks for noting this typographical error from our side, we have since corrected the spelling of statin in the table heading.

It is not correct to state that RCT trials were included is the analysis. main of them were not randomised for the use of statins. in this case, they should considered and evaluated as observational studies

Response: We agree with the reviewers that some studies were not randomised for the use of statin; for instance a study by Negredo in both control and treatment groups, there was the administration of atorvastatin. Therefore we have excluded this study. Following the reviewer's suggestions, we have revised the table to indicate this information about study designs. Such changes are indicated in red throughout the table.

See ref 41 as an example: From the abstract of the study: "Forty-two PLWH aged ≥40 years receiving a protease inhibitor (PI)-based regimen were randomised (1:1) to switch from PI to Raltegravir (n = 20), or to remain on PI (n = 22). After 24 weeks, all patients received atorvastatin 20 mg/day for 48 weeks."

Response: We appreciate this critical comment; the study referred to here is reference 42, a study by Negredo titled “A randomised pilot trial to evaluate the benefit of the concomitant use of atorvastatin and Raltegravir on immunological markers in protease-inhibitor-treated subjects living with HIV”.

After carefully checking the above article following the reviewer's comments, we agree that the above study is irrelevant according to our PICO criteria. One reason is that the atorvastatin was administered after 24 weeks to both control and the Raltegravir group; therefore, we have since removed the study by Negredo from analysis as it is not relevant according to our current PICO criteria. These changes are amended accordingly throughout the manuscript and our supplementary files.

The analysis ". Effect of statin on total cholesterol in HIV patients in ART." is difficult to understand. Statins are very well knwon to reduce cholesterol. what does this analysis add to current knowledge?

Response: We appreciate this important question. However, literature has proven that statins as cholesterols lowering agents still have disparities associated with various forms of statins. Therefore, the main idea for the inclusion of this parameter was to explore in HIV patients who are on ARTs whether there is any drug-to-drug interaction that might overshadow the efficacy of statins on cholesterol. For instance, previous evidence revealed that statin administration should be given with care, noting the drug-to-drug interaction, especially if fluvastatin, lovastatin and simvastatin in ART-treated HIV patients. Although the same study supports the use of pitavastatin, it also indicates limited evidence available to support this.

https://doi.org/10.1016/j.jcte.2017.01.004

the interesting analysis is the one on CD8 and CD4 but it is performed on very few studies.

Response: Thanks for this comment, we agree with reviewers that the analysis was performed using evidence from a few studies, thus limiting the overall interpretation of our findings. Hence, we as researchers need to note this as a main limitation of the current presented evidence to the research community. We have since added a statement in the limitation to highlight this aspect. Refer to page 15, Lines 395-397.

reasons for statin administration in each study is not reported.

Response: Thanks for these comments; although this is an important aspect, we felt it would not add value to the quality of the reporting. However, it’s worth noting that the majority of the studies main aim was to evaluate the effects of statins on lipid profiles, while HIV surrogate markers and immune activation were secondary objectives.

Reviewer 3 Report

The manuscript A systematic review and Meta-Analysis on the impact of statin 2 treatment in HIV patients on Antiretroviral Therapy has solid inclusion/exclusion criteria for the studies chosen for meta-analysis though it might improve the quality of the study if the authors have decided to exclude a few of the studies with the smallest sample numbers (around 20 participants). However, I don't find it to be an obstacle for publishing, the results are clearly presented and the conclusions supported by the results so I find the manuscript suitable for publishing.

Author Response

Comment: The manuscript A systematic review and Meta-Analysis on the impact of statin 2 treatment in HIV patients on Antiretroviral Therapy has solid inclusion/exclusion criteria for the studies chosen for meta-analysis though it might improve the quality of the study if the authors have decided to exclude a few of the studies with the smallest sample numbers (around 20 participants). However, I don't find it to be an obstacle for publishing, the results are clearly presented and the conclusions supported by the results so I find the manuscript suitable for publishing.

Response: Thank you very much for your precious time reviewing our manuscript. We agree with the reviewer's sentiments regarding the small sample size studies, as such studies are underpowered. However, the main idea of the meta-analysis was to combine data from these RCTs despite the small or adequate sample to improve statistical power. Given the limited evidence analysed in this study, excluding studies with a small sample size might have resulted in few studies available for quantitative analysis. Nevertheless, we explored this aspect through sensitivity analysis by removing one study with a small sample size across different outcomes. Refer to page 12, lines 266-279, for detailed information.  

Round 2

Reviewer 2 Report

The change to the table is not correct. Under the column "study design", all RCTs should be indicated as "post-hoc analysis of RCT" as they were not designed to investigate the effect of statins in HIV patients

Author Response

The change to the table is not correct. Under the column "study design", all RCTs should be indicated as "post-hoc analysis of RCT" as they were not designed to investigate the effect of statins in HIV patients.

Response: We appreciate this important comment, as a result, we have revised the study design on the table and included the "post-hoc analysis of RCT"